# An Innovative Formulation Based on Nanostructured Lipid Carriers for Imatinib Delivery: Pre-Formulation, Cellular Uptake and Cytotoxicity Studies

**DOI:** 10.3390/nano12020250

**Published:** 2022-01-13

**Authors:** Evren Gundogdu, Emine-Selin Demir, Meliha Ekinci, Emre Ozgenc, Derya Ilem-Ozdemir, Zeynep Senyigit, Isabel Gonzalez-Alvarez, Marival Bermejo

**Affiliations:** 1Radiopharmacy Department, Faculty of Pharmacy, Ege University, Bornova, Izmir 35100, Turkey; evren.atlihan@gmail.com (E.G.); emine.selin94@gmail.com (E.-S.D.); melihaekinci@ege.edu.tr (M.E.); emreozgenc90@gmail.com (E.O.); deryailem@ege.edu.tr (D.I.-O.); 2Pharmaceutical Technology Department, Faculty of Pharmacy, Izmir Katip Celebi University, Cigli, Izmir 35400, Turkey; zeynepay79@hotmail.com; 3Pharmaceutical Technology Department, Faculty of Pharmacy, Miguel Hernández University, Avenida de la Universidad, 03202 Elche, Spain; mbermejo@umh.es

**Keywords:** imatinib, characterization, nanostructured lipid carrier systems, in vitro release kinetics, cell culture studies

## Abstract

Imatinib (IMT) is a tyrosine kinase enzyme inhibitor and extensively used for the treatment of gastrointestinal stromal tumors (GISTs). A nanostructured lipid carrier system (NLCS) containing IMT was developed by using emulsification–sonication methods. The characterization of the developed formulation was performed in terms of its particle size, polydispersity index (PDI), zeta potential, entrapment efficiency, loading capacity, sterility, syringeability, stability, in vitro release kinetics with mathematical models, cellular uptake studies with flow cytometry, fluorescence microscopy and cytotoxicity for CRL-1739 cells. The particle size, PDI, loading capacity and zeta potential of selected NLCS (F16-IMT) were found to be 96.63 ± 1.87 nm, 0.27 ± 0.15, 96.49 ± 1.46% and −32.7 ± 2.48 mV, respectively. F16-IMT was found to be stable, thermodynamic, sterile and syringeable through an 18 gauze needle. The formulation revealed a Korsmeyer–Peppas drug release model of 53% at 8 h, above 90% of cell viability, 23.61 µM of IC50 and induction of apoptosis in CRL-1739 cell lines. In the future, F16-IMT can be employed to treat GISTs. A small amount of IMT loaded into the NLCSs will be better than IMT alone for therapy for GISTs. Consequently, F16-IMT could prove to be useful for effective GIST treatment.

## 1. Introduction

Recently, cancer has become a common disease, and it can lead to death. It has also affected life expectancy according to the disease level in most developed countries in this century [1,2].

Current therapies for cancer disease include chemotherapy, radiotherapy and surgery. Among them, surgery is an invasive method that can be used in some cases. The techniques of chemotherapy and radiotherapy show multidrug resistance and severe adverse effects in patients. These effects may decrease the achievement of treatment as well as reducing patient compliance. Nanotechnological products have great potential for the delivery of numerous anti-cancer agents. They have demonstrated benefits over traditional chemotherapy [3]. 

Imatinib (IMT) is a tyrosine kinase enzyme inhibitor and is effective in the treatment of gastrointestinal stromal tumors (GISTs) [4,5]. It has some undesired adverse effects such as cardiac, pulmonary and hepatic toxicity [6]. The benefits associated with the particular properties of nanostructured lipid carrier systems (NLCSs) include tumor-specific drug deposition, efficient pharmacokinetics and pharmacodynamics on account of the therapeutic agent, enhanced internalization and intracellular transport and decreased biodistribution, resulting in the mitigation of the harmful consequences of anti-tumor treatments. 

NLCSs are the second generation of lipid nanoparticles and have the ability to enable the insertion of a liquid lipid into a solid lipid matrix. This forms a space in the crystal lattice. As a result, the drug molecule has a better accommodation in the NLCSs. In addition, these systems improve the drug loading capacity, reduce drug expulsion during storage, provide high drug encapsulation efficiency and cope with the drawback of drug expulsion from the lipid phase [7,8,9]. 

The potential usage of NLCSs to deliver anticancer drugs via oral, intravenous, pulmonary and intraperitoneal routes of administration has been expressed in several studies [10,11,12]. Herein, IMT-loaded NLCSs were produced by using emulsification and sonication methods—procedures that have relatively low costs and exclude any use of organic solvents, and, in this case, all the components are GRAS (generally recognized as safe). Different types of surfactants, solid and liquid lipids and solvents were studied to obtain stable formulations. Furthermore, preparation conditions such as the stirring rate and time and sonication were evaluated. The developed formulation was observed in terms of particle size, PDI and zeta potential measurements, thermal analysis, entrapment efficiency, loading capacity, microbiological analysis, syringeability study, stability and in vitro release studies. In addition, cellular uptake and cytotoxicity studies of NLCSs containing IMT were performed by using CRL-1739 cells.

## 2. Materials and Methods

### 2.1. Materials 

Gelucire pellet and lipoid derivatives were obtained from Gattefosse Sas/Saulieu, France and LIPOID GmbH/Frankfurt, Germany, respectively. Span 80 and Tween 80 were purchased from Sigma-Aldrich Chemie/Hamburg, Germany. Imatinib (Gleevec^®^) was obtained from Novartis (Basel, Switzerland). CRL-173 was purchased from ATCC (Maryland, MD, USA). All other reagents were in our research laboratory.

### 2.2. Solubility Study

The solubility of IMT in several phases was observed. One milligram of IMT was put into vials containing 1 mL of different phases. The systems were mixed at 25 ± 1 °C for 24 h. After that, the samples were centrifuged at 1000 rpm for 5 min and filtered to remove insoluble IMT by using 0.45 µm membrane filters. The amount of IMT was analyzed by HPLC (Thermo Scientific Accela PDA Detector/IET|International Equipment Trading Ltd., Illinois, IL, USA).

### 2.3. HPLC Analysis of IMT

The samples were analyzed by using an HPLC system. The mixture of acetonitrile and triethylene amine (60:40, *v/v*) was used as the mobile phase at a flow rate of 1.0 mL/min. Fifty microliters of samples were injected into the C_18_ column (250 mm × 4.6 mm, LiChroCART, Munich, Germany), and UV detection was performed at 266 nm. The calibration curve was drawn, and linearity was provided over 1–100 μg/mL concentration with good linearity (r^2^  =  0.998).

### 2.4. Preparation of NLCS

The preparation of NLCS were performed by using emulsification–sonication methods [10,11,12,13]. Oleic acid and compritol, oleic acid and gelucire derivatives (Gelucire 48/16 and Gelucire 43/01 pellets) were used as the lipid phase. Surfactants comprised Span 80, Tween 80 and lipoid derivatives. The lipid phases were melted in a water bath (70–80 °C). One milligram of IMT was added to the lipid phase. An aqueous solution containing water and a water:acetone:ethanol mixture with surfactants was prepared and heated to the same temperature as the lipid phase. All systems were mixed at 10,000 rpm for 10 min using an Ultra-Turrax blender (IKA^®^-Werke GmbH & Co., Staufen, Germany) to obtain emulsification. The sonication was carried out at 500 W and 20 kHz in changing 20 s cycles for 15 min by using a Vibracell tip sonicator (IKA^®^-Werke GmbH & Co., Staufen, Germany) and then brought to room temperature.

### 2.5. Physicochemical Characterization of NLCS

#### 2.5.1. Particle Size, Polydispersity Index and Zeta Potential

The formulations were evaluated with a Malvern Zetasizer (Malvern Nano ZS 90, Malvern, UK) in a particle size range of 3–1000 nm at room temperature with an angle of 173° to determine the aggregate formation particle size and polydispersity index. Samples were diluted with distilled water (pH = 7). The zeta potential of formulations was measured at 40 V/cm using a DTS 1060C zeta cuvette (Malvern, UK) at 25 °C, with 5 mS/cm of conductivity and a dielectric constant of 78.5. Measurements were made in triplicates and results were exhibited as mean value ± SD.

#### 2.5.2. Syringeability Study

The syringeability study was performed by using a TA-XT Texture Analyzer (Stable Micro Systems Ltd., Godalming, UK) at the 30 kg load cell stage. A syringe probe (Universal Syringe Rig Ltd., Godalming, UK) and pressure mode were used. The ejection time of the 5 mL formulation under selected conditions was recorded as the syringeability time. 

#### 2.5.3. Scanning Electron Microscopy (SEM)

The size and surface properties of the formulations were examined under a high vacuum on a scanning electron microscope (Philips XL 30S FEG model, Leuven, Belgium) by the Thermo Scientific Apreo S brand. For this purpose, the samples were first coated with the Leica EMACE 600 brand coating device with 80% gold and 20% palladium to a thickness of 7 nm. The coating was made under a vacuum of 5 × 10^−4^ mBar. The scanning of the coated samples was implemented at a magnification range of ×50,000 and increased voltage conditions of 5 kV.

#### 2.5.4. Differential Scanning Calorimetry (DSC)

The IMT powder, free formulations and IMT formulations were evaluated by differential scanning calorimetry (DSC Q2000 V24.11 Build 124, Perkin Elmer, Massachusetts, ABD, MA, USA) at 20–300 °C for aluminum recrystallization, and a 10 °C temperature rise and 50 mL/min nitrogen flow rate was used for recrystallization and disintegration product formation. 

#### 2.5.5. Entrapment Efficiency (EE) and Loading Capacity (LC)

The entrapment efficiency (EE) and loading capacity (LC) of IMT in NLCSs were determined by using a dialysis bag (Spectrum Ltd., Shanghai, China) with 8000–12,000 Da of molecular weight [9]. For that, 1 mL of NLCSs containing IMT was put into dialysis bags. The samples were ultra-centrifuged at 5000 rpm at room temperature and filtered by using cellulose nitrate membrane. The concentration of IMT was analyzed by HPLC. The calculation of EE and LC was conducted by using the following equations [12,14]:EE (%) = ((W_total drug_ − W_free drug_)/W_total drug_) × 100 
LC (%) = ((W_total drug_ − W_free drug_)/W_total formulation_) × 100 
where
W_total drug_ = The total amount of drug;
W_free drug_ = The amount of drug in the supernatant;
W_total formulation_ = The amount of formulation.

#### 2.5.6. Stability Study

The stability at 5 ± 3 °C, 25 ± 5 °C and 60 ± 5% relative humidity (RH), 40 ± 5 °C and 75 ± 5% (RH) of the free and IMT loaded formulations was observed at certain time intervals for 6 months. The particle size, PDI, zeta potential and visual appearance of formulations were evaluated. All values were compared statistically for the initial and time interval values.

### 2.6. In Vitro Drug Release Study

The in vitro release of IMT from NLCSs was fulfilled by using a dialysis bag in pH 7.4 phosphate buffer at 37 °C under sink conditions. The magnetic stirring was adjusted to 100 rpm. In total, 250 μL of samples were withdrawn for 8 h for HPLC analysis. Then, 250 μL of buffer solution was added to the medium to complete the release volume. All measurements were performed in triplicate. The IMT release percentages were calculated by peak areas in the chromatograms. Several kinetic models were tested, and the best kinetic model of the IMT release profile was chosen according to the determination coefficient (r^2^) [15]. 

The zero order model is as follows:Q = k·t + Q_0_

where Q is the amount (%) of drug released at time t, Q_0_ is the starting value of Q, and k is the rate constant.

The Korsmeyer–Peppas model is as follows:Q = k·t_n_

where n is the release exponent.

The Hixon–Crowell model is as follows:m = ^3^√W_0_ − ^3^√W 
where m is the amount of drug released at time t, ^3^√W_0_ is the cube root of the initial concentration, and ^3^√W is the cube root of the concentration at time t [16].

### 2.7. In Vitro Cytotoxicity Studies

The in vitro cytotoxicity of NLCSs containing IMT on CRL-1739 that mimic gastric adenocarcinoma was studied. The cell line was cultured in DMEM supported with 10% fetal bovine serum, 0.5 mg/mL glutamine/penicillin streptomycin at 37 °C and 5% CO_2_. The cytotoxicity of IMT-loaded NLCSs with different concentrations (0.05 mg/mL, 0.1 mg/mL, 0.2 mg/mL, 0.4 mg/mL, 0.5 mg/mL, 1 mg/mL, 2 mg/mL) and pure IMT (dissolved in cell medium) was determined in CRL-1739 using the 3-[4,5-dimethylthiazole-2-yl]-2,5-diphenyltetrazolium bromide (MTT) assay. Ninety-six well flat-bottom plates and 1 × 10^6^ cells/well in 1 mL DMEM were used for the seeding procedure. The NLCSs formulations and IMT solutions at various concentrations were then added into the 96 well plates for 48 h. IMT was solubilized in DMEM (1 mg/mL) and incubated at concentrations of 0.05 mg/mL, 0.1 mg/mL, 0.2 mg/mL, 0.4 mg/mL, 0.5 mg/mL, 1 mg/mL and 2 mg/mL. Then, 100 μL of MTT solution (5 mg/mL) was put into each well and incubated at 37 °C. The culture medium was removed. Then, 200 μL of dimethysulfoxide was added to obtain the dissolution of the MTT formazan crystals. The absorbance was read at 570 nm using a microplate reader. Furthermore, the “GraphPad Prism” and “ne-site total binding” algorithm programs were used to calculate 50% inhibition of viability (IC_50_) [17,18]. 

### 2.8. Microbiological Analysis of Formulations

#### 2.8.1. Sterility Test

All formulations (F9-IMT, F15-IMT, F16-IMT, F19-IMT, F20-IMT and F21-IMT) were incubated with Thioglycollate Broth and Tryptic Soy Broth medium at 37  ±  2  °C in the vials. The vials were controlled in terms of their visibility during the sterility test. 

#### 2.8.2. Pyrogenicity Test

The pyrogenicity test was conducted by using Pyrotell Gel-Clot formulation (Associates of Cape Cod Incorporated, Falmouth, MA, USA).

### 2.9. Cellular Uptake Study Flow Cytometry Cell Apoptosis Analysis and Fluorescence Microscopy

The cellular uptake study of IMT solution, NLCS (F16) and IMT-loaded NLCS (F16-IMT) was performed by using flow cytometry. The control group comprised untreated cells and fresh culture medium without test compounds. CRL-1739 cells were seeded to a 5 × 10^4^ cell density in 24-well plates. NLCS (F16), IMT solution and IMT-loaded NLCSs (F16-IMT) with doses of 0.05 mg/mL and 2 mg/mL were applied to the cells for 24 h. After the administration of formulations, cells were washed with PBS three times. The cells were trypsinized and suspended in fresh PBS with 10% fetal bovine serum. 

#### 2.9.1. Flow Cytometry Cell Apoptosis Analysis

The suspended formulations were incubated with 5 μL of Annexin V-FITC and 5 μL of propidium iodide for 15 min at room temperature in the dark. The cell suspensions were analyzed by using Guava easy Cyte in terms of cellular uptake.

#### 2.9.2. Fluorescence Microscopy

After treatment with the formulations, CRL-1739 cells were fixed and stained with Hoechst 33342 (blue). The cellular uptake of formulations was performed with a Nikon eclipse fluorescence microscope (Nikon, Japan) at a magnification of 1000×. The stained cell nuclei were evaluated.

### 2.10. Statistical Analysis

The statistical analysis was performed by using an analysis of variance (ANOVA) program. Differences between results were considered statistically significant when the *p*-values were less than 0.05. Results were expressed as mean ± standard deviation (SD).

## 3. Results

### 3.1. Solubility Studies

Solubility studies were carried out in water, surfactants and lipids, which can be used in the preparation of NLCSs. In many studies, it is desirable to have a solubility of more than 10 mg/mL of the active substance, especially for ease of formulation development. The solubility of IMT with different vehicles has been reported in the literature [19,20]. As a result of solubility studies, it was observed that the solubility of IMT was also found to be above 10 mg/mL in water, ethanol, gelucire 43/01, gelucire 48/16, compritol 888, lipoid S 75 and lipoid S 100 (Table 1).

### 3.2. Preformulation Studies

#### Preparation of NLCS

The preparation of NLCS was accomplished to select a convenient proportion of lipid, surfactant, solvents, stirring rate, stirring time and sonication conditions to develop stable NLCS. Different formulations containing different ratios of lipids and surfactants have been evaluated (Table 2). Oleic acid, gelucire derivatives and compritol 888 were used as liquid lipid and solid lipids, respectively. In the preparation process, IMT was added to the melted lipid phase. Water and a water:acetone:ethanol mixture with surfactants were prepared as aqueous solutions. Span 80, Tween 80 and lipoid derivatives were used as surfactants. Especially, Span 80 and Tween 80 are a polysorbate class of amphiphiles. They have good emulsification ability for lipid mixtures. and their combination with the lipid phase causes the formation of nanostructures and leads to the stabilization of NLCS. 

### 3.3. Physicochemical Characterization of NLCS

#### Particle Size, Zeta Potential and Polydispersity Index

The physicochemical characterization parameters of developed NLCSs are summarized in Table 3. The measured parameters influence the formulation’s stability and their future interactions with cells and tissues. According to the obtained results, the particle size, PDI and zeta potential of developed NLCSs ranged from 75 to 231 nm, 0.08 to 0.924 and −10 to −55 mV, respectively (Table 3). The formulations were monitored over 30 days of storage at 4 °C, 25 °C (RH 60%) and 40 °C (RH 75%) as recommended by the ICH guideline (Table 4). The particle size for some formulations (F9, F15, F16, F19, F20, F21) was in accordance with the former results (Table 3) and confirmed the reproducibility of the preparation procedure. The other formulations indicating minus values (−) were removed because of their particle size and PDI variation, phase separation or agglomeration of the NLCS particles.

### 3.4. Preparation and Characterization NLCSs Containing IMT

The NLCSs containing IMT (F9-IMT, F15-IMT, F16-IMT, F19-IMT, F20-IMT, F21-IMT) were prepared as described with free formulations (F9, F15, F16, F19, F20, F21). The addition of IMT was carried out in heated lipid phases. The characterization of NLCSs containing IMT was performed in terms of particle size, zeta potential and PDI. 

The particle size of NLCSs is important since it influences the rate and extent of drug release from the system. The drug release is enhanced by a smaller particle size, which results in a larger surface area for drug release between oil and water phases. The average particle size of NLCSs containing IMT was found to be between 82.43 ± 1.26 and 162.3 ± 6.28 nm, with a PDI of 0.15 ± 0.034 and 0.37 ± 0.14 (Table 5) indicating uniformity in particle size distribution. The zeta potential for NLCSs containing IMT showed that they have a negative moiety with a zeta potential between (−28.7 ± 2.26 mV) and (−41.12 ± 3.73 mV) (Table 5). Additionally, statistically significant differences between the free and IMT formulations were seen for the particle size, PDI and zeta potential values (Table 4 and Table 5). 

### 3.5. Syringeability

Syringeability is significant for the administration of parenteral preparations into the body using a syringe and needle. The syringeability results showed that prepared NLCS was easily syringeable through an 18 gauze needle. The syringeability time recorded for formulations was found to be between 7.5 and 10 s. The highest syringe force was observed in F21 and F19 (Table 6). This could be based on the different particle size and zeta potential of the systems and results in an altered resistance to flow [21,22]. 

### 3.6. Scanning Electron Microscopy (SEM)

The SEM images were recorded to investigate the surface phase and size properties of all formulations (Figure 1) and revealed that the particles have a spherical shape. Furthermore, the particle size results obtained by SEM analysis (Figure 1) were consistent with the particle size measurement results obtained with the Malvern Zetasizer (Table 5). Some clusters were encountered in the images, which might be connected with the shrinkage of NLCSs during drying or the concentration of dispersion medium.

### 3.7. Differential Scanning Calorimetry (DSC)

DSC presents the polymorphic changes in formulations and is helpful for the evaluation of their thermodynamic properties and possible drug–formulation interactions [23]. The DSC thermograms of IMT and all formulations are shown in Figure 2, Figure 3 and Figure 4. The melting point of IMT has been reported in the literature to be 222–224 °C [24]. According to DSC analysis results, the melting temperature of IMT obtained from the study was found to be consistent with the value stated in the literature. The DSC thermogram displays the disappearance of an IMT peak in the formulations, which shows that IMT is surrounded inside the NLCSs. 

### 3.8. Entrapment Efficiency (EE) and Loading Capacity (LC)

The amounts of IMT entrapped in the six optimized formulations and the loading capacity of NLCSs containing IMT are listed in Table 7. All of them have high EE and LC values. This result demonstrates that the composition of developed formulations is suitable for the delivery of IMT.

### 3.9. Stability Study

A stability study was performed for free and IMT-loaded formulations at 5 ± 3 °C, 25 ± 5 °C and 60 ± 5% relative humidity (RH), 40 ± 5 °C and 75 ± 5% (RH) over 6 months. While each NLCS formulation containing IMT (F9-IMT, F15-IMT, F16-IMT, F19-IMT, F20-IMT, F21-IMT) showed statistically significant differences in terms of particle size, no statistically significant difference was observed in these particle sizes over 6 months. On the other hand, the particle sizes were found to be in accordance with expected values for NLCSs (below 200 nm) (Figure 5). Therefore, IMT did not change the structural situation of NLCSs depending on the time. PDI values did not exceed 0.5 for the formulations (Figure 5), and all formulations presented a homogenous distribution during the storage period in spite of the low increment in size as for some formulations. In addition, zeta potential results demonstrated that statistically significant differences between the formulations were not observed (Figure 5) under storage conditions and periods. As a result, the optimal formulations were chosen according to particle size and zeta potential, and they ensured physicochemical stability for up to 6 months.

### 3.10. In Vitro Drug Release Study

The IMT release profiles of NLCSs with different compositions are shown in Figure 6, Figure 7 and Figure 8. No burst effect was observed during in vitro drug release study. The reason could be the high encapsulation of IMT to lipid drug delivery systems, which could prevent the rapid drug release. While the highest release was observed in F16-IMT and F19-IMT, at around 53% and 57%, respectively, a low release was obtained in F21-IMT after 8 h. Various mathematical equations were performed to define the kinetic model of IMT release results. F16-IMT and F19-IMT formulations showed suitable kinetic models (zero order, Korsmeyer–Peppas and Hixon–Crowell models) according to the r^2^ values (r^2^ > 0.95) at phosphate buffer. Particularly, the Korsmeyer–Peppas model is found to be appropriate for F16-IMT formulation. The remaining NLCSs (F9-IMT, F15-IMT, F20-IMT, F21-IMT) are much smaller for in vitro IMT release kinetic models, and no changes were found in comparison to the rest of the models (*p* < 0.05). 

### 3.11. In Vitro Cytotoxicity Studies 

The in vitro cytotoxicity studies of IMT solution and NLCSs containing IMT were performed with CRL-1739 cells and are presented in Figure 9. While F16-IMT demonstrated above 90% cell viability, the other formulations exhibited above 80% cell viability, and the IMT solution showed 70.5% cell viability after 48 h treatment, suggesting that the NLCS system—especially F16-IMT—is more available than the IMT solution form in terms of cytotoxicity. Additionally, IMT-NLCSs indicated dose-dependent cytotoxicity since the cell viability ratio changed with the alteration of IMT concentration ranging from 0.05 mg/mL to 2 mg/mL. IC_50_ values of all formulations were calculated by using GraphPad Prism and found to be between 8.53 ± 2.7 and 24.54 ± 1.44 µM for CRL-1739 cells (Figure 10). These values suggested that the IMT concentrations applied in CRL-1739 cells were not likely to cause toxicity, and the administration of the formulations can be considered to correspond to desired drug delivery systems. 

### 3.12. Microbiological Analysis of Formulations

The microbiological analysis was carried out in the absence and growth of microorganisms in the vials containing all formulations visibly and clearly. A gel-clot test was also performed, and the results revealed that all formulations were pyogenic-free and the formulations which were prepared at the laboratory scale and under aseptic conditions can be used for parenteral administration. 

### 3.13. Cellular Uptake Studies with Flow Cytometry and Fluorescence Microscopy

Flow cytometry was used to study F16, IMT solution and F16-IMT with the presence of 0.05 and 2 mg/mL of intracellular uptake by CRL-1739 cells. Although there are differences in terms of cellular uptake for F16, IMT solution and F16-IMT, all formulations were taken up by the cells. The cells that are administered with IMT solution showed lower fluorescence intensity than cells administered with F16 and F16-IMT. The results (Figure 11 and Figure 12) revealed that IMT has a low penetration ability. Furthermore, it can be expected that the low biological activity of IMT in solution form will be shown with respect to the poor diffusion of IMT into the cells. 

IMT location and distribution inside CRL-1739 cells were investigated by using the Hoechst 33342 and fluorescence microscopy. In Figure 11, IMT solution exhibited low fluorescence when compared to F16 and F16-IMT. While IMT was encountered inside the nucleus, which resulted in the purple fluorescent spots in Figure 11 and was enough to indicate a cytotoxic effect for cells, IMT penetration into the cell nucleus was not observed for F16-IMT, and IMT was distributed in the cell cytoplasm. This is caused by the presence of NLCS in F16-IMT. 

Figure 12 shows the apoptotic effects of IMT solutions (0.05 and 2 mg/mL), F16, F16-IMT (0.05 and 2 mg/mL) on CRL-1739 cells. The number of cells in the apoptosis of F16 and F16-IMT is higher than that of IMT solution. In terms of evaluating the different concentrations of IMT in solution and NLCS forms, CRL-1739 cells begin to undergo necrosis, and the ratio of dead cells reached 23.26% with an increase in concentration from 0.05 to 2 mg/mL of IMT. This study proved that drug nano-carrier systems containing IMT can be based on the reduction of cell viability by the induction of apoptosis in CRL-1739 cell lines.

## 4. Discussion

NLCSs are lipid-based nano drug carrier systems and they comprise liquid lipid, solid lipid, surfactants and aqueous phases. In this study, many formulation contents were tested during the NLCSs preparation procedure, and some formulations were eliminated because of phase separation and creamy formation. F16-IMT was selected as an ideal formulation according to its good characterization and stability results. This formulation comprised compritol 888, oleic acid, lipoid S 75 and a water:acetone:ethanol mixture. Oleic acid and compritol 888 were used as liquid and solid lipids, respectively. Lipoid S 75 leads to the good emulsification of the lipid mixture and provides stabilization of NLCS. Hydrophilic solvents such as water, ethanol, propanol, butanol, pentanol and hexanol are used as the aqueous phase. The rapid distribution of the lipid phase into the aqueous phase is obtained by its critical behavior in NLCS formation [25,26,27]. Furthermore, their low toxicity, owing to the higher solubilizing capacity, reduced surface tension and facilitation of forming properties, is helpful to prepare NLCSs with small particle sizes [28,29]. The preparation of NLCSs was achieved by using the emulsification–sonication method. The facilitation of the production process, short production time, homogeneous and desired particle size and the obtaining of a stable formulation were carried out thanks to this method [14,15,29,30,31]. 

Gupta et al. [32] prepared imatinib-loaded nanostructured lipid carriers (IMT-NLC) by the hot homogenization method. While their formulations contain Precirol ATO 5, Tween^®^ 20, Labrafil M 1944 CS and lecithin, our formulations include compritol 888, oleic acid, lipoid S 75 and a water:acetone:ethanol mixture. In addition, Gupta et al. [32] encountered difficulties in the application of temperature during the preparation process because of the protein structure of IMT. The F16-IMT developed herein is prepared by emulsification and sonication methods, and these methods eliminate the difficulty of hot homogenization.

The characterization of formulations was performed in terms of particle size, PDI and zeta potential values. Particle size, zeta potential and PDI are significant parameters for the characterization of NLCSs and are effective in the interaction of formulations with the biological system [33]. The desired particle size, zeta potential and PDI values were achieved by 500 W and 20 kHz for between 5–15 min at 5000–10,000 rpm. In the presence of a surfactant, a high zeta potential value, small particle size and desired PDI value were accomplished during the preparation process and stability studies. In some studies, the results revealed that nano carrier drug delivery systems with particle sizes of 300 nm and below 300 nm are more suitable for pharmaceutical applications because they are recognized by erythrocytes and other cells and maintained higher drug release properties compared to nano carrier drug delivery systems with large sized particles [33,34]. The particle sizes of all NLCSs were found to be below 300 nm and available for pharmaceutical administration according to the literature [35,36]. In this study, high and negative zeta potential values were obtained due to usage of the surfactant concentration in the preparation process. The PDI values of the all formulations were below 0.5, which is indicative of monodispersed formulations, with uniform diameters of nanoparticle populations [25,29,30,37]. While some formulations have a PDI value below 0.5, the others do not meet this requirement. These PDI values below 0.5 also suggest a uniform functionalization process.

IMT is a tyrosine kinase enzyme inhibitor with a specific affinity for Bcr-Abl, PDGFR and c-Kit. It has been shown to be highly effective in the treatment of GIST, characterized by the over-expression of Bcr-Abl. In total, 400 mg/day of IMT is used in conventional therapy. However, it has low aqueous solubility, an insufficient pharmacokinetic profile and undesired cardiotoxic effect [34]. NLCSs also allow the encapsulation of the drug, with the capability to increase its aqueous solubility. The circulation time greatly improves its potential chemo-therapeutic effect. NLCSs are samples of the carrier system and capable of enhancing the solubility, permeability, cellular uptake and bioavailability of encapsulated drugs. Based on these considerations, and as targeted in the study, F16-IMT was prepared by using a smaller amount of IMT (1 mg of IMT) than the amount of IMT in commercial formulation. The characterization of NLCSs containing IMT was performed in terms of particle size, zeta potential and PDI values. As mentioned in previous studies, a narrow size distribution proves that Ostwald ripening can also be seen in the formulations [38,39]. Gupta et al. [32] obtained that IMT-NLC exhibited a particle size of 148.80 ± 1.37 nm, PDI of 0.191 ± 0.017 and of zeta potential of −23.0 ± 1.5 mV. The differences between the characterization parameter values are caused by variations in formulation contents.

The zeta potential demonstrates the net electrostatic charge on the particle surface and is a crucial factor to evaluate the stability of colloidal systems. Particles with negative or positive zeta potential values give information about the stability of NLCSs [26,36,40,41]. According to the zeta potential results, F16-IMT had negative zeta potential values and saved the values during storage conditions.

The EE and LC of drug formulations are important properties to achieve the accurate administration of the drug delivery system. Obtaining desired values for EE and LC prevents dose-dependent side effects, and thus patient compliance is improved [30,32]. In this study, high EE and LC values were obtained for developed NLCSs that contained a lower IMT amount than commercial formulation (Table 6).

A stability study is necessary to show the physicochemical characteristics of formulations and that they are maintained over time, since the degradation of NLCSs could influence their potential as effective drug delivery systems [42]. The obtained results exhibited that the appearance and characterization parameters of the formulations did not change during the stability study (Figure 5).

The formulations were monitored for 30-day storage at 4 °C, 25 °C (RH 60%) and 40 °C (RH 75%) as recommended by the ICH guideline, considering the particle size and PDI (Table 4). The particle size values for some formulations (F9, F15, F16, F19, F20, F21) were shown to be in accordance with former results (Table 3) and confirmed the reproducibility of the preparation procedure. The other formulations showing negative values (−) were removed because of their particle size and PDI variation and phase separation or the agglomeration of the NLCS particles.

According to the literature, a high amount of drug delivery was achieved using lipid-based formulations with particle sizes of 120–300 nm. Therefore, NLCSs were found to be appropriate for carrying the drugs to the desired site thanks to their particle sizes. The value of PDI is an indicator of the homogeneity of particle size distribution in a lipid-based dispersion, and it is under 0.5 in the monodisperse formulations [42,43,44,45,46]. The size distribution of all formulations might be defined as broad due to their PDI values, which are higher than 0.2. Hence, the developed NLCSs were found to be polydisperse samples.

NLCSs have an amorphous structure in their matrix. This structure allows a larger amount of drug to be encapsulated compared to other lipid-based systems [46,47]. Herein, a high IMT entrapment efficiency was observed in the developed formulation (F16-IMT) (Table 7). This provides clear evidence for the greater drug encapsulation tendency of NLCSs [47,48,49,50]. In addition, the results demonstrate that the composition of the developed formulation is suitable for the delivery of IMT.

The IMT release profiles of NLCSs were evaluated in different media (Figure 6, Figure 7 and Figure 8). In NLCSs formulations, the lipid matrix can erode slowly, and entrapped drugs remain inside the core of the NLCs and present prolonged release [46]. According to in vitro release study results, prolonged release was observed for some formulations. In addition, the amount of surfactant affected the release of IMT from NLCSs. When the surfactant concentration is increased, drug release increased. This may be related to the surfactant-induced increase in drug solubility in the aqueous phase, which would facilitate its diffusion from the lipid matrix to the outer aqueous environment. Mathematical modeling has shown the expression of the release mechanism of drugs from the carrier systems that lead to the release profiles. Various mathematical equations were performed to define the kinetic model of IMT release results [51]. From these different models, F16-IMT and F19-IMT formulations showed suitable kinetic models (zero order, Korsmeyer–Peppas and Hixon–Crowell models) according to the r^2^ values (r^2^ > 0.95) for phosphate buffer. Particularly, the Korsmeyer–Peppas model is found to be appropriate for F16-IMT formulation. The most commonly used kinetic model in drug release studies, especially for lipid nanoparticles, is the Korsmeyer–Peppas model, which defines the drug transport through Fickian or non-Fickian transports [52,53]. According to kinetic model calculations, Fickian transport was found to be suitable for IMT release of F16-IMT since there is an efflux of IMT from high amount to low amount depending on time [53], and IMT release can be controlled with the erosion mechanism [54]. The remaining NLCSs (F9-IMT, F15-IMT, F20-IMT, F21-IMT) are much smaller for in vitro IMT release kinetic models, and no changes were found in comparison to the rest of the models (*p* < 0.05).

The charge of particles affects the cytotoxicity of formulations. NLCSs with negatively charged particles present less cytotoxicity than positively charged particles. This is generally reconciled with cell membrane disruption and consequent cell death [44]. The oleic acid and compritol 888 mixture were used to prepare NLCSs which have a negative zeta potential value [45,46]. When nanosystems are in close vicinity to a cell, interactions can be obtained between the nanosystem and the cell. This leads to the membrane wrapping of the nanosystem followed by cellular uptake. Cytotoxicity study helps to understand how the nanosystem influences cell viability so that their undesirable properties can be avoided. Cellular uptake and cytotoxicity studies also cause nanoparticles to move into the clinical arena. The cellular uptake and cytotoxicity results indicated that the increased cellular uptake of IMT and reduction of IMT toxicity on CRL-1739 were obtained with NLCSs and were found to be in accordance with previously reported studies [55,56,57].

## 5. Conclusions

This work defines preformulation and formulation studies that resulted in the new development of NLCSs containing IMT. Characterizations of formulations were performed in terms of particle size, PDI, zeta potential, entrapment efficiency, loading capacity, stability, sterility and syringeability studies, and all results prove the availability of F16-IMT for intravenous administration. Structural and morphological analyses (DSC and SEM) confirmed that F16-IMT was successfully prepared by emulsion and sonication techniques. The best release for IMT was obtained from F16-IMT, and the Korsmeyer–Peppas model was found to be a good kinetic model for the formulation. F16-IMT represented a lower toxic effect on CRL-1739 cells that mimic GIST when compared to IMT solutions. Furthermore, F16-IMT showed an enhanced anticancer effect and was involved in apoptosis induction. Accordingly, a reduced dose will provide an enhanced anti-cancer effect on cancer patients. Consequently, less adverse effects and reduced exposure of the body will be observed. Therefore, the F16-IMT may be considered suitable as a therapeutic agent for GIST after carrying out studies on animal models.

## Figures and Tables

**Figure 1 nanomaterials-12-00250-f001:**
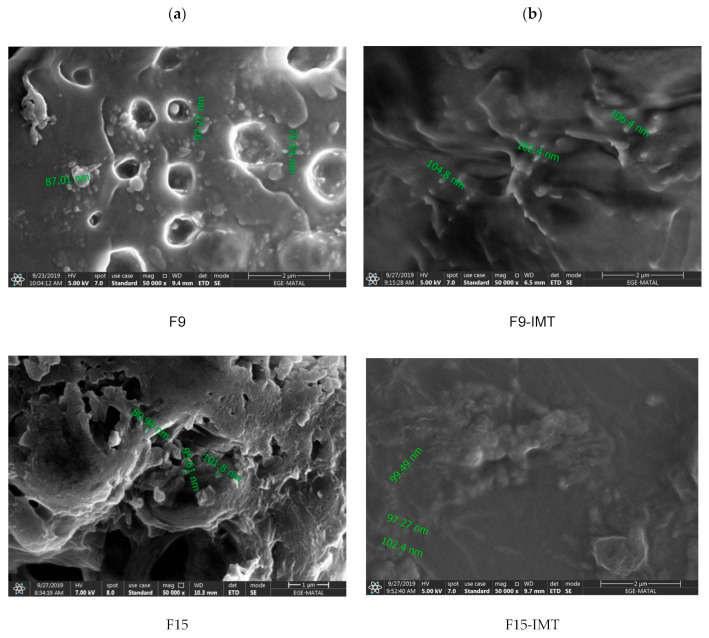
(**a**) SEM images of free formulations, (**b**) SEM images of formulations containing IMT.

**Figure 2 nanomaterials-12-00250-f002:**
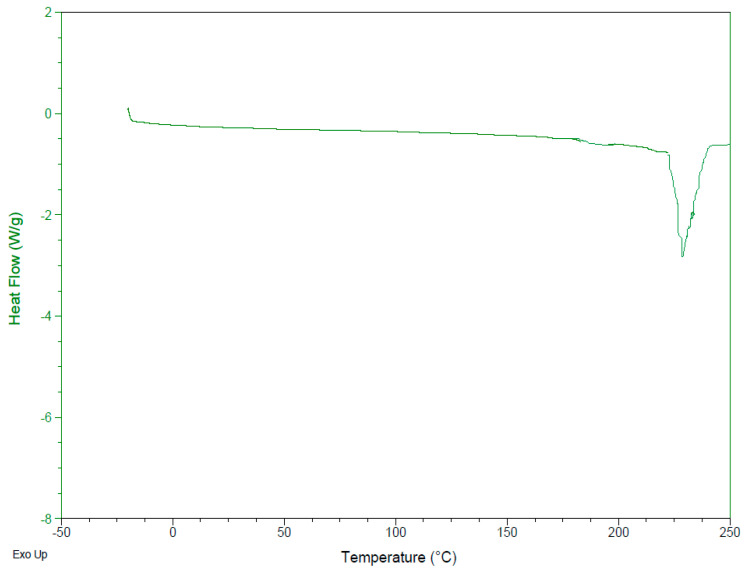
DSC thermogram of IMT.

**Figure 3 nanomaterials-12-00250-f003:**
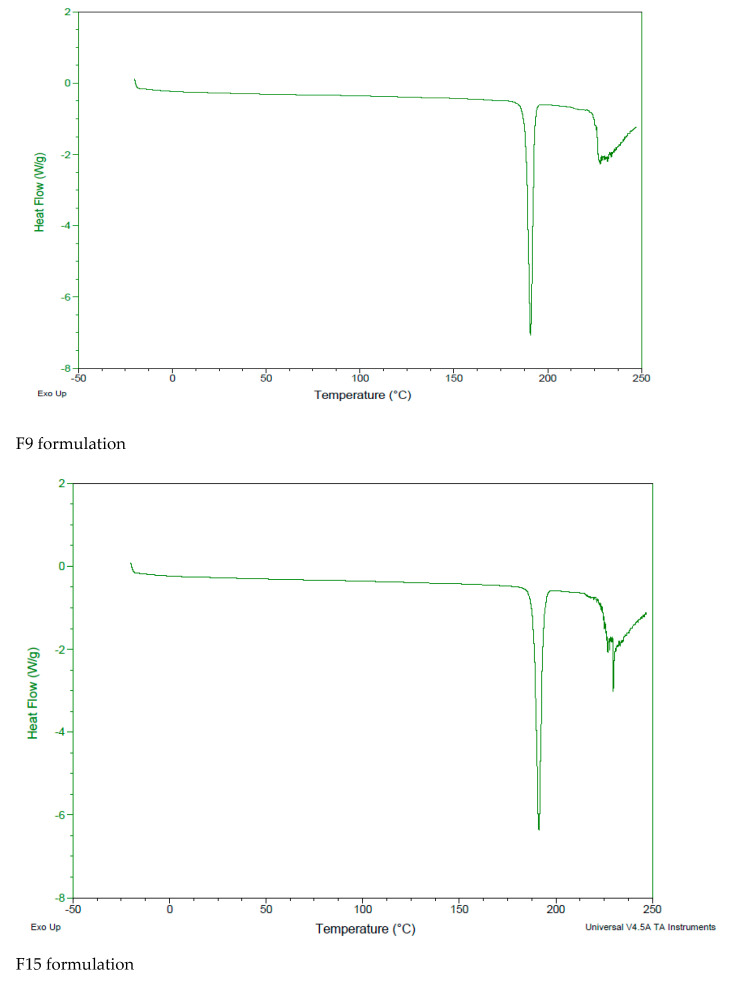
DSC thermogram of F9, F15, F16, F19, F20 and F21.

**Figure 4 nanomaterials-12-00250-f004:**
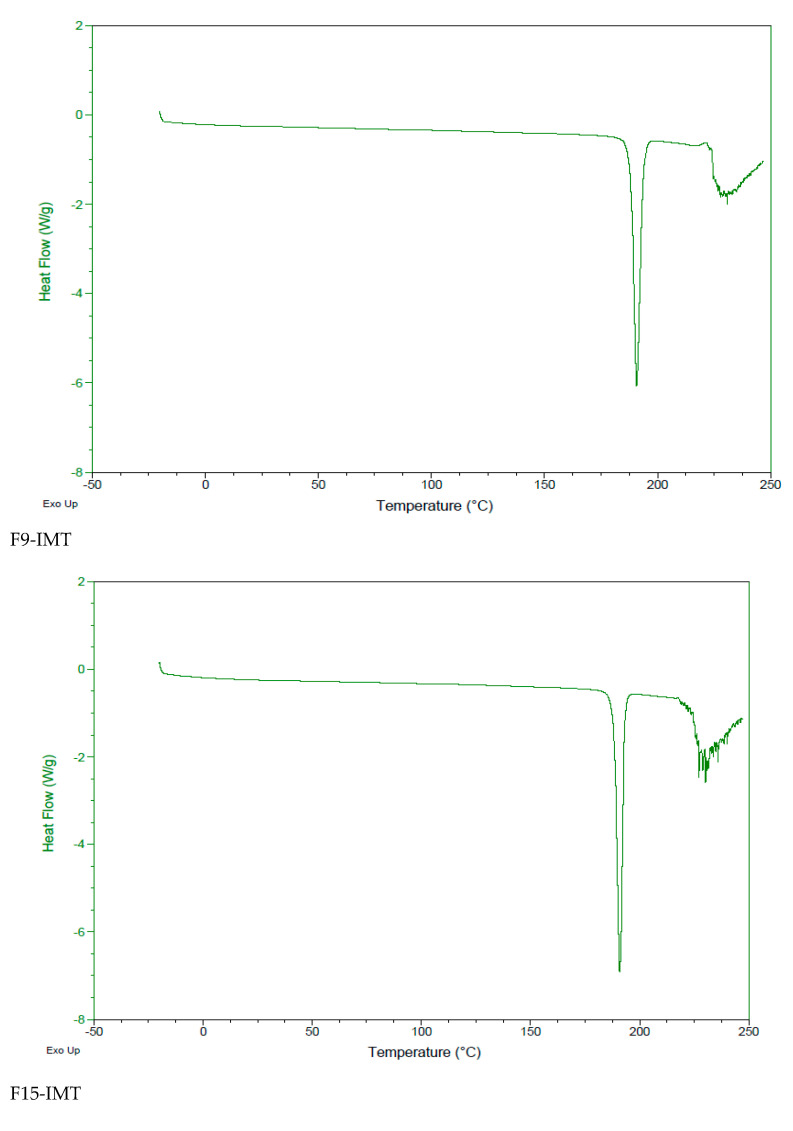
DSC thermogram of F9-IMT, F15-IMT, F16-IMT, F19-IMT, F20-IMT and F21-IMT.

**Figure 5 nanomaterials-12-00250-f005:**
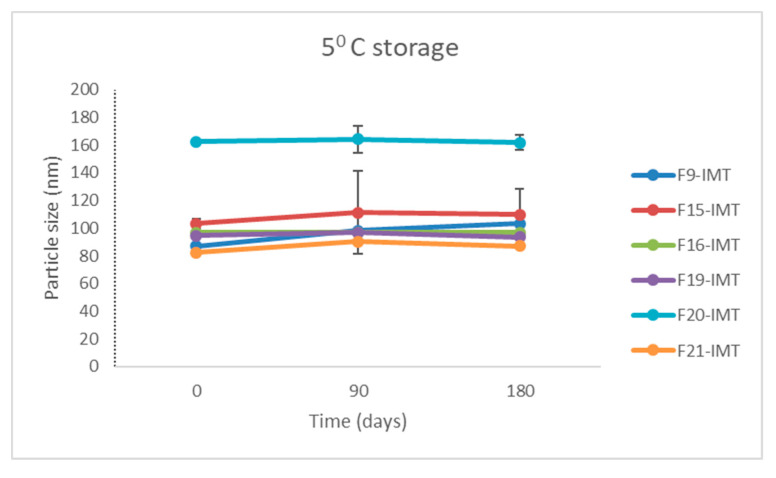
Stability of NLCSs considering particle size, PDI and zeta potential values for 6 month storage.

**Figure 6 nanomaterials-12-00250-f006:**
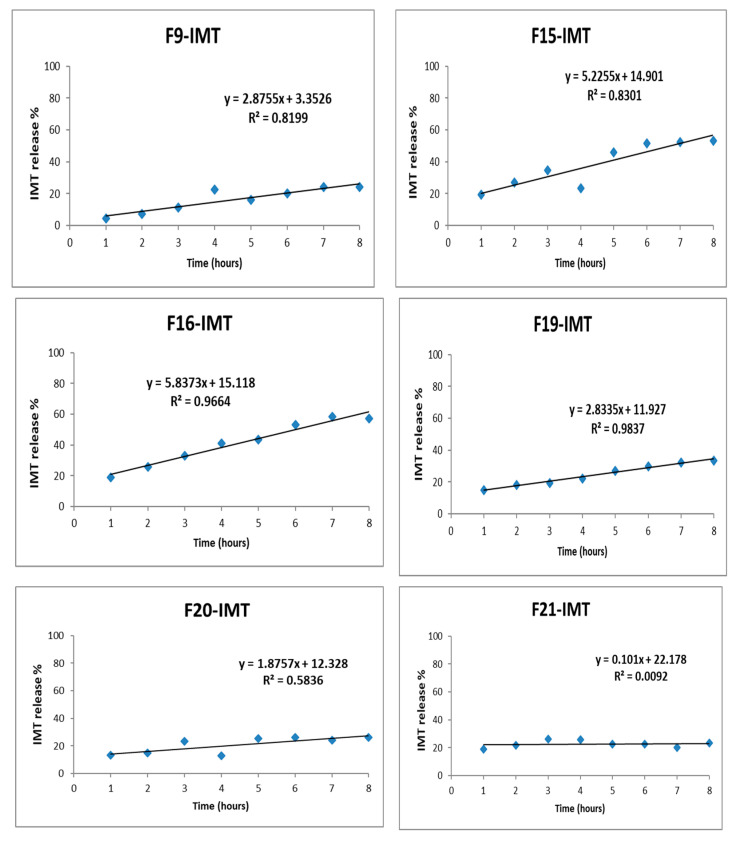
In vitro release profile of IMT from developed formulations with zero order kinetic modeling.

**Figure 7 nanomaterials-12-00250-f007:**
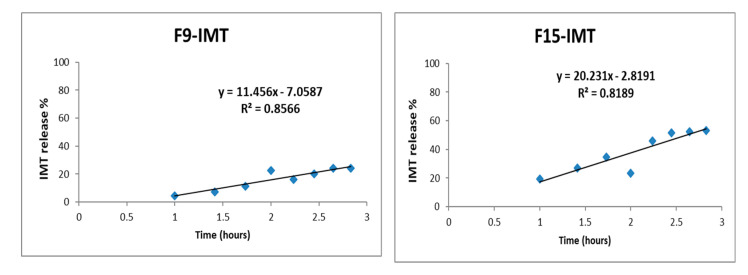
In vitro release profile of IMT from developed formulations with Korsmeyer–Peppas kinetic modeling.

**Figure 8 nanomaterials-12-00250-f008:**
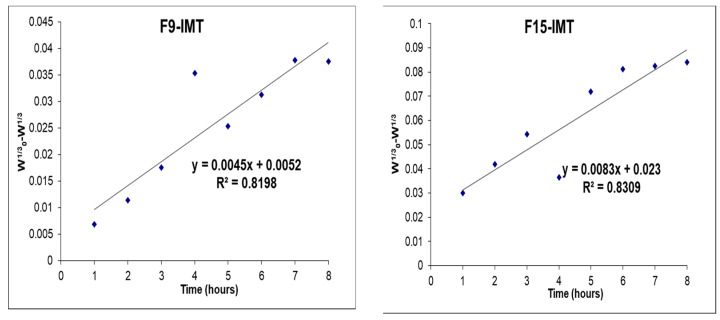
In vitro release profile of IMT from developed formulations with Hixon–Crowell kinetic modeling.

**Figure 9 nanomaterials-12-00250-f009:**
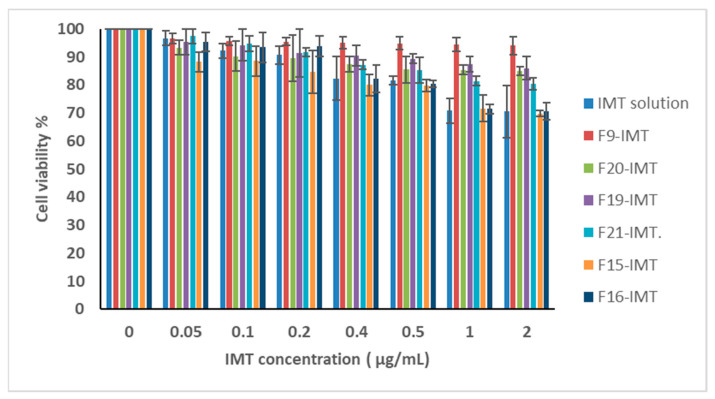
In vitro cytotoxicity of IMT solution and NLCSs with different IMT concentration µ.

**Figure 10 nanomaterials-12-00250-f010:**
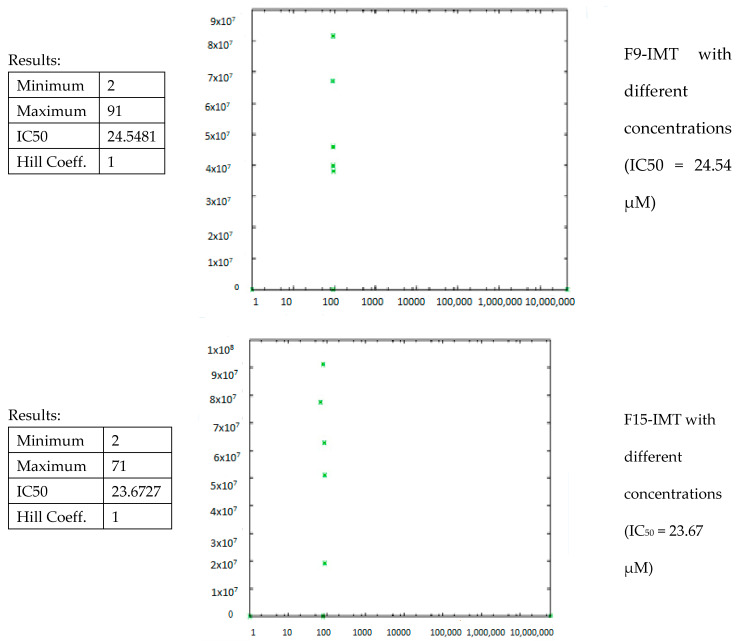
IC_50_ values of formulations.

**Figure 11 nanomaterials-12-00250-f011:**
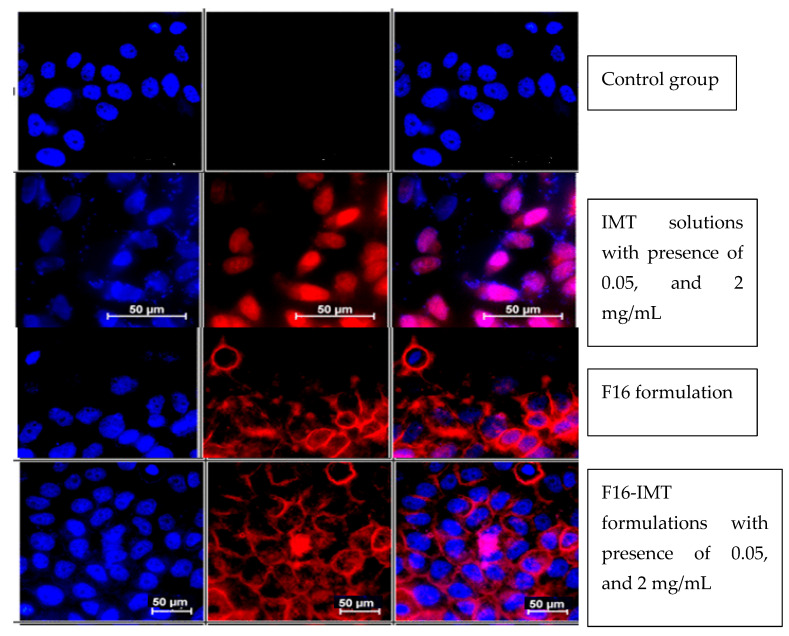
Qualitative analysis of IMT distribution in CRL-1739 cells treated with IMT solutions (0.05, and 2 mg/mL), F16, F16-IMT (0.05, and 2 mg/mL).

**Figure 12 nanomaterials-12-00250-f012:**
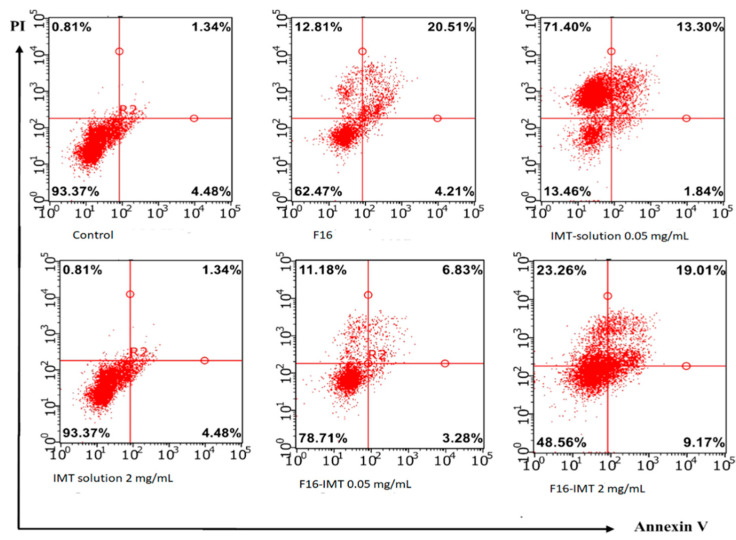
Apoptotic effects of IMT solutions (0.05 and 2 mg/mL), F16, F16-IMT (0.05 and 2 mg/mL) on CRL-1739 cells. Values are presented as the mean ± SD (n = 3).

**Table 1 nanomaterials-12-00250-t001:** The solubility of IMT in different vehicles.

Phases	Amount of IMT (mg/mL) ± SD
Water	0.0082 ± 0.0011
Span 80	3.61 ± 0.63
Tween 80	4.23 ± 0.45
Ethanol	10.19 ± 1.03
Oleic acid	8.80 ± 0.65
Gelucire 43/01	10.62 ± 1.03
Gelucire 48/16	11.06 ± 2.3
Lipoid S 75	12.23 ± 3.4
Lipoid S 100	10.41 ± 1.21
Compritol 888	10.29 ± 1.01

**Table 2 nanomaterials-12-00250-t002:** Composition of the NLCS formulations and preparation conditions.

(F1–F3) Contents and Preparation Conditions							
**Formulations**	**Gelucire 48/16 Pellets (%)**	**Oleic Acid (%)**	**Tween 80 (%)**	**Water (%)**	**Stirring Rate (rpm)**	**Stirring Time (min)**	**Sonication**
F1	33.33	33.33	22.22	11.11	10.000	5	500 W and 20 kHz
F2	25	25	37.5	12.5	10.000	5	500 W and 20 kHz
F3	16.67	16.67	50	16.67	10.000	5	500 W and 20 kHz
(F4–F6) contents and preparation conditions							
**Formulations**	**Gelucire 43/01 Pellets (%)**	**Oleic Acid (%)**	**Span 80 (%)**	**Water (%)**	**Stirring Rate (rpm)**	**Stirring Time (min)**	**Sonication**
F4	37.5	37.5	12.5	12.5	5000	10	500 W and 20 kHz
F5	25	25	25	25	5000	10	500 W and 20 kHz
F6	28.57	28.57	14.28	28.57	5000	10	500 W and 20 kHz
(F7–F9) contents and preparation conditions							
**Formulations**	**Gelucire 48/16 Pellets (%)**	**Oleic Acid (%)**	**Span 80 (%)**	**Water (%)**	**Stirring Rate (rpm)**	**Stirring Time (min)**	**Sonication**
F7	28.57	28.57	28.57	14.28	10.000	15	500 W and 20 kHz
F8	28.57	14.28	28.57	28.57	10.000	15	500 W and 20 kHz
F9	25	25	25	25	10.000	15	500 W and 20 kHz
F10 content and preparation condition							
**Formulation**	**Compritol 888 (%)**	**Oleic Acid (%)**	**Span 80 (%)**	**Water (%)**	**Stirring Rate (rpm)**	**Stirring Time (min)**	**Sonication**
F10	66.67	13.33	6.66	13.35	10.000	5	500 W and 20 kHz
(F11–F18) contents and preparation conditions							
**Formulations**	**Compritol 888 (%)**	**Oleic Acid (%)**	**Lipoid S 75 (%)**	**Water:Acetone: ikaika Ethanol (5:2.5:2.5 *v/v*) %**	**Stirring Rate (rpm)**	**Stirring Time (min)**	**Sonication**
F11	30	30	30	10	10.000	5	500 W and 20 kHz
F12	33.33	33.33	22.22	11.11	10.000	5	500 W and 20 kHz
F13	25	25	37.63	12.37	10.000	5	500 W and 20 kHz
F14	28.57	28.57	28.57	14.28	10.000	5	500 W and 20 kHz
F15	16.67	16.67	50	16.67	10.000	5	500 W and 20 kHz
F16	20	20	40	20	10.000	5	500 W and 20 kHz
F17	25	25	25	25	10.000	5	500 W and 20 kHz
F18	28.57	14.28	28.57	28.57	10.000	5	500 W and 20 kHz
(F19–F25) contents and preparation conditions							
**Formulations**	**Gelucire 48/16 Pellets (%)**	**Oleic Acid (%)**	**Lipoid S 100 (%)**	**Water:Acetone: ikaika Ethanol (5:2.5:2.5 *v/v*) %**	**Stirring Rate (rpm)**	**Stirring Time (min)**	**Sonication**
F19	33.33	33.33	33.33	11.11	10.000	5	500 W and 20 kHz
F20	25	25	37.5	12.5	10.000	5	500 W and 20 kHz
F21	28.57	28.57	28.57	14.28	10.000	5	500 W and 20 kHz
F22	33.33	33.33	16.67	16.67	10.000	5	500 W and 20 kHz
F23	16.67	16.67	50	16.67	10.000	15	500 W and 20 kHz
F24	20	20	40	20	10.000	15	500 W and 20 kHz
F25	25	25	25	25	10.000	15	500 W and 20 kHz

**Table 3 nanomaterials-12-00250-t003:** The particle size, PDI and zeta potential values of the formulations.

Formulations	Particle Size (nm ± SD)	PDI	Zeta Potential (mV ± SD)
F1	174.1 ± 28.78	0.733 ± 0.081	−24.2 ± 3.40
F2	99.30 ± 40.90	0.398 ± 0.043	−18.5 ± 2.85
F3	19.01 ± 0.89	0.345 ± 0.016	−10.1 ± 2.78
F4	231.01 ± 9.28	0.365 ± 0.114	−50.4 ± 2.11
F5	181.7 ± 5.48	0.357 ± 0.029	−55.2 ± 1.50
F6	197.6 ± 5.10	0.33 ± 0.05	−52.0 ± 0.27
F7	177.4 ± 11.38	0.52 ± 0.12	−42.1 ± 2.05
F8	172.8 ± 9.35	0.40 ± 0.09	−36.1 ± 0.42
F9	75.2 ± 2.50	0.408 ± 0.11	−35.7 ± 1.18
F10	178.2 ± 14.56	0.44 ± 0.06	−38.1 ± 1.46
F11	129.33 ± 4.37	0.44 ± 0.05	−33.56 ± 3.52
F12	59.75 ± 1.138	0.92 ± 0.04	−33.56 ± 0.72
F13	128.13 ± 0.76	0.08 ± 0.06	−32.56 ± 6.16
F14	165.1 ± 6.76	0.19 ± 0.03	−37.60 ± 1.19
F15	102.86 ± 0.76	0.15 ± 0.004	−37.43 ± 2.41
F16	95.25 ± 2.77	0.10 ± 0.02	−3130 ± 0.68
F17	122.5 ± 2.17	0.11 ± 0.05	−24.56 ± 1.2
F18	148.28 ± 83.25	0.23 ± 0.02	−30.13 ± 0.42
F19	83.32 ± 1.92	0.43 ± 0.04	−34.9 ± 0.64
F20	149.4 ± 2.95	0.13 ± 0.04	−39.98 ± 0.21
F21	78.28 ± 0.99	0.32 ± 0.01	−38.1 ± 0.38
F22	124.31 ± 31.91	0.22 ± 0.04	−25.76 ± 3.51
F23	153.1 ± 0.15	0.12 ± 0.03	−12.6 ± 5.06
F24	151.1 ± 3.91	0.15 ± 0.05	−18.0 ± 0.55
F25	135.5 ± 2.11	0.197 ± 0.01	−12.93 ± 0.49

**Table 4 nanomaterials-12-00250-t004:** The particle size and PDI values of the formulations over 30 days.

	0 Day	15 Days			30 Days		
	25 ± 2 °C	5 ± 3 °C	25 ± 2 °C, %60 RH	40 ± 2 °C, %75 RH	5 ± 3 °C	25 ± 2 °C, %60 RH	40 ± 2 °C, %75 RH
F1	174.1	259.2	549.7	729.3	-	-	-
SD	28.78	16.00	11.3	8.83	-	-	-
PDI	0.73	0.65	0.66	0.61	-	-	-
F2	99.30	521.1	1241	1202	-	-	-
SD	40.90	54.3	25.5	14.22	-	-	-
PDI	0.39	0.57	0.92	0.91	-	-	-
F3	19.00	400.2	324.2	295.3	-	-	-
SD	0.89	112.4	75.75	20.2	-	-	-
PDI	0.34	0.56	0.63	0.57	-	-	-
F4	231.0	1518	1703	1394	-	-	-
SD	9.28	1205	895	567	-	-	-
PDI	0.36	0.87	0.91	0.86	-	-	-
F5	181.7	208.7	146.6	520.2	258.43	157.2	158.96
SD	5.48	8.60	3.06	10.16	9.22	8.62	13.3
PDI	0.35	0.30	0.19	0.28	0.25	0.15	0.29
F6	197.6	341.2	230.0	174.6	-	-	-
SD	5.10	127.7	8.35	3.26	-	-	-
PDI	0.33	0.44	0.36	0.18	-	-	-
F7	177.4	207.1	175.1	285.7	-	-	-
SD	11.38	13.66	4.85	31.96	-	-	-
PDI	0.51	0.39	0.44	0.41	-	-	-
F8	172.8	302.9	520.0	624.8	-	-	-
SD	9.35	85.51	248.3	202.4	-	-	-
PDI	0.40	0.47	0.57	0.68	-	-	-
F9	75.2	76.1	75.8	81.4	82.73	75.13	87.63
SD	2.50	10.1	4.21	4.64	3.32	7.52	1.08
PDI	0.41	0.45	0.38	0.41	0.37	0.39	0.41
F10	178.2	417.5	316.7	289.1	-	-	-
SD	14.56	10.4	8.73	49.91	-	-	-
PDI	0.42	0.38	0.64	0.53	-	-	-
F11	129.33	285.4	311.7	364.5	-	-	-
SD	4.36	13.32	41.98	62.03	-	-	-
PDI	0.43	0.51	0.75	0.67	-	-	-
F12	59.75	288.4	181.9	309.7	951.3	87	406.96
SD	1.14	12.7	61.33	23.21	240.82	5.51	55.6
PDI	0.92	0.82	0.85	0.61	0.613	1	0.46
F13	128.13	161.2	390.5	601.4	-	-	-
SD	0.75	4.454	20.38	15.54	-	-	-
PDI	0.08	0.11	0.16	0.41	-	-	-
F14	165.1	192.1	406.7	678.7	263.26	502.7	216.86
SD	6.76	8.58	4.71	67.00	25.39	60.21	36.53
PDI	0.19	0.097	0.33	0.670	0.2	0.21	1
F15	102.86	121.3	123.7	103.8	125.03	120.97	104.73
SD	0.76	1.51	3.38	6.86	5.11	1.95	5.73
PDI	0.15	0.08	0.17	0.71	0.07	0.032	0.71
F16	95.256	102.8	127.9	130.4	108	130.23	113.23
SD	2.77	1.33	3.68	9.34	0.79	5.22	6.45
PDI	0.10	0.11	0.17	0.19	0.042	0.07	0.17
F17	122.5	149.2	157.3	432.6	145.5	147.73	847.37
SD	2.17	3.50	2.55	33.53	6.36	4.22	33.45
PDI	0.10	0.17	0.21	0.15	0.11	0.16	0.35
F18	148.28	505.5	448.8	518.8	-	-	-
SD	83.25	191.7	247.9	28.53	-	-	-
PDI	0.23	0.60	0.54	0.52	-	-	-
F19	83.32	65.9	45.08	49.91	63.33	45.26	52.73
SD	1.92	2.44	0.612	3.56	3.611	0.87	7.06
PDI	0.43	0.50	0.28	0.48	0.24	0.21	0.51
F20	149.4	157.1	139.3	97.40	182.87	195.73	169.68
SD	2.95	4.22	10.86	1.16	1.46	2.06	1.23
PDI	0.13	0.13	0.30	0.34	0.05	0.36	0.38
F21	78.28	82.3	79.76	68.97	78.43	59.17	60.72
SD	0.99	0.63	1.28	1.11	4.23	1.47	0.64
PDI	0.32	0.41	0.43	0.38	0.41	0.29	0.27
F22	124.31	993.7	410.9	199.3	-	-	-
SD	31.91	633.0	84.76	20.54	-	-	-
PDI	0.22	0.83	0.51	0.75	-	-	-
F23	153.1	201.7	350.1	122.9	265.73	258.5	126.13
SD	0.15	7.68	11.03	1.95	9.06	12.93	20.79
PDI	0.12	0.24	0.41	0.37	0.04	0.95	0.50
F24	151.1	210.5	327.7	61.45	212.53	55.6	156.73
SD	3.915	20.42	33.75	17.37	13.45	31.1	5.84
PDI	0.15	0.34	0.35	0.64	0.24	0.20	0.49

**Table 5 nanomaterials-12-00250-t005:** The particle size, PDI and zeta potential values of the formulations with and without IMT.

Formulations	Particle Size (nm ± SD)	PDI	Zeta Potential (mV ± SD)
F9-IMT	86.93 ± 2.73	0.37 ± 0.14	−38.13 ± 2.16
F15-IMT	103.16 ± 3.64	0.27 ± 0.16	−28.7 ± 2.26
F16-IMT	96.63 ± 1.87	0.27 ± 0.15	−32.7 ± 2.48
F19-IMT	94.43 ± 2.28	0.35 ± 0.12	−37.9 ± 2.91
F20-IMT	162.3 ± 6.28	0.15 ± 0.034	−41.12 ± 3.73
F21-IMT	82.43 ± 1.26	0.34 ± 0.21	−37.66 ± 2.66

**Table 6 nanomaterials-12-00250-t006:** Syringeability time and syringe force values of NLCSs.

Formulations	Syringeability Time (s)	Syringe Force (g)
F9-IMT	7.5 ± 1.72	234.79 ± 7.46
F15-IMT	6.51 ± 1.24	155.03 ± 3.55
F16-IMT	7.74 ± 0.69	186.74 ± 5.10
F19-IMT	8.73 ± 2.86	271.58 ± 18.83
F20-IMT	9.21 ± 1.47	233.0 ± 8.58
F21-IMT	9.26 ± 6.03	286.26 ± 27.0

**Table 7 nanomaterials-12-00250-t007:** Entrapment efficiency and loading capacity of F9-IMT, F15-IMT, F16-IMT, F19-IMT, F20-IMT and F21-IMT (*p* > 0.05).

Formulations	Entrapment Efficiency (µg ± SD)	Loading Capacity (% ± SD)
F9-IMT	951.74 ± 3.36	95.96 ± 1.8
F15-IMT	932.96 ± 2.48	95.19 ± 0.85
F16-IMT	967.07 ± 11.67	96.49 ± 1.46
F19-IMT	851.59± 4.73	86.31 ± 3.08
F20-IMT	961.19 ± 5.66	93.76 ± 3.17
F21-IMT	962.49 ± 25.48	96.42 ± 1.48

## Data Availability

This study excluded this statement.

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
