# Peer review of "An Innovative Formulation Based on Nanostructured Lipid Carriers for Imatinib Delivery: Pre-Formulation, Cellular Uptake and Cytotoxicity Studies"

_nanomaterials, 2022, doi:10.3390/nano12020250_

Round 1

Reviewer 1 Report

This work may be of value for readers especially those working in the similar direction. This work however needs susbtantial improvement..

Although I made comments in the attached file, I started reading the manuscript but quality of presentation in terms of English and figure quality is poor. It was almost impossible to see the figures and hence could not reach to any conclusion. I would suggest authors to present data in a concise way instead of supplying everything just to over-load the manuscript with un-relevant data. Unless, you submit the improved manuscript, it would be difficult for anyone to reach any conclusion

Author Response

Dear reviewer,

Thank you for your valuable comments. The quality of presentation in terms of English was checked by native speaker and figure quality was improved. The recommended publications were added to manuscript.

attach the final version with changes.

Reviewer 2 Report

This manuscript entitled “An innovative formulation based on nanostructured lipid carriers for imatinib delivery: Pre-formulation, cellular uptake and cytotoxicity studies” written by Gundogdu et al. is aimed to give a complete elucidation of the potential usage of NLCSs, produced by using emulsification and sonication methods, to deliver anticancer drugs.

Potentially the MS could be interesting but seems not so well organized and and is hard to follow.

The quality of the figures is very low. some of them seem to be a screenshot (see DSC figure 2 and others). Table 2 is 4 pages, also table 4. Many typos are detected and a native speaker should correct the MS. Probably degree of melting is used as "melting temperature" or "melting point". References are not matching the guidelines of the journal. Paragraphs are not numbered. The rules for significant figures of the values and errors in the tables need corrections. SEM figures and other figures are not readable. 

The MS needs a deep revision.

For these reasons, the MS is hard to follow and I recommend rejection.

Author Response

Thank you for your valuable comments. The quality of presentation in terms of English was checked by native speaker and figure quality was improved. The rules for significant figures of the values and errors in the tables were corrected. The degree of melting was changed as "melting temperature" or "melting point". The corrections was made on references according to guidelines of journal.

The final manuscript is attached

Round 2

Reviewer 1 Report

Figure 10 is still blur and difficult to understand

I would suggest authors to give thermograms as supplemental information

I suggest authors to go thru their manuscript thoroughly to address typos and grammatical errors

Author Response

Thank you for your valuable comments. The quality of presentation in terms of English was checked by native speaker and figure quality was improved. The recommended publications were added to manuscript.

Reviewer 2 Report

The work deserves to be published because some interesting data are found.

Unfortunately, the MS is overloaded by data and seems to be an infinite list of information. The discussion seems to be un-related, poor and for these reasons not interesting.

I suggest to re-write the discussion in an appropriate way. Concise and with focus on the aim of the MS.

Again, extensive editing of English language and style is required.

Author Response

Thank you for your valuable comments. The quality of presentation in terms of English was checked by native speaker and figure quality was improved. The rules for significant figures of the values and errors in the tables were corrected. The degree of melting was changed as "melting temperature" or "melting point". The corrections was made on references according to guidelines of journal.

Round 3

Reviewer 1 Report

I am satisfied by the authors response and manuscript has been improved significantly...

Reviewer 2 Report

Figures need revision.